# A Bacterium Derived from the Ovary of the Black Soldier Fly (*Hermetia illucens*) Attract Oviposition of the Host

**DOI:** 10.3390/biology14091107

**Published:** 2025-08-22

**Authors:** Muyang He, Yi Wang, Wenxuan Xu, Guohui Yu, Xun Yan

**Affiliations:** 1Innovative Institute for Plant Health, Zhongkai University of Agriculture and Engineering, Guangzhou 510225, China; hemuyang@zhku.edu.cn (M.H.); qwangyi0108@163.com (Y.W.); yqddxwx@163.com (W.X.); 2Key Laboratory of Green Prevention and Control on Fruits and Vegetables in South China, Zhongkai University of Agriculture and Engineering, Ministry of Agriculture and Rural Affairs, Guangzhou 510225, China

**Keywords:** *Hermetia illucens*, *Serratia marcescens*, dimethyl disulfide, oviposition, genome analysis

## Abstract

As the world’s population keeps growing, we urgently need better ways to use our limited resources. The black soldier fly offers a promising solution—this insect can turn food waste and agricultural leftovers into valuable protein. However, making this process work on a large industrial scale faces challenges because the flies do not always lay eggs efficiently in controlled environments. Our research discovered a special helper: bacteria living inside these flies that naturally encourage egg-laying. We identified one particular bacterium from the reproductive system of the flies that produces a chemical signal. This signal attracts mother flies to lay eggs in treated areas. We also mapped the entire genetic blueprint of this bacterium to understand how it creates this egg-laying signal. This discovery is promising to develop natural, bacteria-based products to boost the reproduction of the flies without chemicals. By increasing egg production, we can grow more black soldier flies to recycle more food waste into animal feed and fertilizer.

## 1. Introduction

We are confronted with unprecedented challenges, as the escalating pressures of population growth impose increasingly stringent demands on resource and food utilization. According to statistical data, the global population is projected to surpass 10 billion by the mid-21st century [1]. Effectively harnessing limited resources has become an urgent predicament that necessitates immediate attention. The utilization of waste through resource insects presents a viable solution to address this pressing issue. The black soldier fly (BSF, *Hermetia illucens*), as an exemplar resource insect, exhibits the ability to convert decaying organic matter into its own proteins through larval feeding [2,3]. This distinctive characteristic makes it an optimal tool for efficient resource recycling, as well as for applications in animal and aquatic feed [4].

The efficient mass cultivation of BSF indoors has long presented a significant challenge within the industry [5]. In recent years, numerous researchers have devoted their efforts to enhancing the comprehension of breeding practices for BSF. Investigations have unveiled that various factors, including light, temperature, nutrition, and substrate composition, can significantly impact the reproductive capacity of BSF [6,7,8]. Moreover, previous studies indicate that microorganisms also exert an influence on egg-laying behavior in BSF. For instance, Zheng et al. demonstrated that bacteria isolated from insects competing for resources with BSF larvae inhibit the egg deposition of BSF adults [9]. However, this research only focused on microbial-mediated repellent effects in BSF, while other potential interaction mechanisms remain underexplored.

By recognizing odor signals produced or induced by microorganisms in their surroundings, insects exhibit corresponding attraction or avoidance behaviors. For instance, *Pseudomonas aeruginosa* commonly found on plant surfaces produces 2-aminoacetophenone, a volatile substance similar to “grape taste,” thereby enhancing food attraction for *Mediterranean fusca*, houseflies, and fruit flies [10]. *Drosophila* can recognize geosmin produced by *Penicillium* and consequently display avoidance behavior [11]. However, there is currently limited research regarding the attraction behavior of microorganisms towards BSF [12]. Exploring the use of microorganisms to attract the BSF to lay eggs holds promising potential.

In this study, we sought to improve the oviposition efficiency of BSF via utilization of its microbiota. we isolated a bacterium from the ovaries of BSF that exhibits a strong attraction for BSF oviposition. The active odorant compounds responsible for this attraction were identified through gas chromatography–mass spectrometry (GC-MS) analysis and behavioral assays. Additionally, genome sequencing of the strain was performed to delineate its genetic repertoire and characterize functional potential through comprehensive bioinformatic annotation.

## 2. Materials and Methods

### 2.1. BSF Colony

Rearing protocols for BSF larvae were adapted from Zhao et al. [13], with modifications to dietary substrates and environmental parameters as detailed below. BSF larvae were procured from Hainan Fuya Environmental Protection Technology Development Co., Ltd., Hainan, China. The larvae were reared in a light incubator maintained at 28 °C, 70% humidity, and a 16:8 light–dark cycle. The feed consisted of a mixture of bran, chicken feed (Huachu Trading Co., Ltd., Guangzhou, China. Formulated primarily with maize, soybean, and wheat; detailed formulation parameters are provided in the manufacturer’s technical specifications.) and cornmeal in a volumetric ratio of 5:3:2. The larvae were reared until the pre-pupal stage, at which point they were transferred to black plastic boxes (5 cm × 10 cm × 5 cm) for pupation. Upon emergence, the adult flies were moved to a cage (60 cm × 60 cm × 90 cm) housed indoors under controlled conditions of 28 °C and 70% humidity. Then, 5% (*w*/*v*) honey water was provided in the cage to supplement adult nutrition, facilitating mating. Egg-laying boards were placed inside the cage to collect eggs, which were subsequently incubated to restart the life cycle. Same feed was administered throughout the experimental period.

### 2.2. Stress-Induced Oviposition of BSF

Fertilized female BSF individuals were immobilized with fine-tipped forceps and positioned in a sterile Petri dish. Surface decontamination was performed by washing in 1.5% (*w*/*v*) sodium hypochlorite solution for 20 s. The following procedures were performed under microscopic guidance: (1) separate head from the thoracic segment; (2) separate thorax and abdomen; (3) upon head separation, stress-induced oviposition responses initiated. Separation of thorax and abdomen triggered increased oviposition output. All procedures were conducted within a laminar flow cabinet to maintain aseptic conditions.

### 2.3. Isolation and Identification of Bacteria

R_2_A medium was prepared consisting of yeast powder (0.5 g), casein peptone (0.5 g), beef peptone (0.5 g), hydrolyzed casein (0.5 g), glucose (0.5 g), starch (0.5 g), sodium pyruvate (0.3 g), dipotassium hydrogen phosphate (K_2_HPO_4_, 0.3 g), magnesium sulfate heptahydrate (MgSO_4_·7H_2_O, 0.024 g), and agar powder (15 g) in 1 L of distilled water. The pH was adjusted to 7.2 using 1 mol/L NaOH, and the medium was sterilized at 115 °C for 20 min. Eggs under stress-induced oviposition conditions and 0, 8, 24 h postpartum were collected. 0.02 g Eggs from BSF collected were thoroughly homogenized, mixed with 1 mL of sterile water, and 100 μL of this suspension was evenly spread on the R_2_A medium. Cultures were incubated at 26 °C, 30 °C, and 37 °C, with three replicates at each temperature. Distinct colonies based on morphological differences were streaked and purified. Purified strains were selected as single colonies and cultured overnight in test tubes containing 5 mL of LB broth for preservation and DNA extraction.

DNA extraction from bacterial samples was conducted following the manufacturer’s instructions provided with the DNA extraction kit (Nanjing Nuoweizan Biotechnology Co., Ltd. Nanjing, China). Subsequently, PCR amplification was performed using universal primers targeting the 16S rDNA region: primer 27F (5′-AGAGTTTGATCMTGGCTCAG-3′) and primer 1492R (5′-TACCTTGTTACGACTT-3′). The reaction mixture consisted of 2 μL template DNA, 2 μL of each primer at a concentration of 10 μmol/L, 25 μL Taq DNA polymerase, and 19 μL ddH_2_O. The PCR cycling conditions were as follows: initial denaturation at 95 °C for 3 min; followed by 30 cycles of denaturation at 95 °C for 15 s, annealing at 55 °C for 20 s, and extension at 72 °C for 90 s; final extension at 72 °C for 10 min; and a hold at 12 °C.

The purified DNA was obtained using the FastPure Gel DNA Extraction Mini Kit. The concentration and quality of the purified DNA were assessed via 1% agarose gel electrophoresis and spectrophotometric analysis, followed by ligation with a T-vector for transformation. The ligation reaction mixture (5 μL) comprised the following: recovered DNA (2 μL), 5× TA/Blunt-Zero Cloning Mix (from the 5 minTM TA/Blunt-Zero Cloning Kit) (1 μL), and ddH_2_O (2 μL). The ligation reaction was performed at 26 °C for 5 min using a PCR thermocycler. Post-reaction, 50 μL of competent cells were added to the mixture and incubated on ice for 30 min. Following transformation, the cells were resuspended in 200 μL of LB medium and incubated at 37 °C with shaking at 180 rpm for 1 h. The transformed cells were then plated on LB agar supplemented with ampicillin and cultured overnight at 37 °C. Positive clones were selected and sequenced by Bioengineering Co., Ltd., Shanghai, China. Sequence comparisons were conducted using the NCBI database, and phylogenetic trees were constructed using MEGA7 software.

### 2.4. Whole-Genome Sequencing and Analysis of Serratia marcescens Hei101

The genomic DNA was extracted by using the Cetyltrimethyl Ammonium Bromide (CTAB) method with minor modification, and the DNA concentration, quality, and integrity were then determined by using a Qubit Flurometer (Invitrogen, CA, USA) and a NanoDrop Spectrophotometer (Thermo Scientific, CA, USA). Sequencing libraries were generated using the TruSeq DNA Sample Preparation Kit (Illumina, CA, USA) and the Template Prep Kit (Pacific Biosciences, CA, USA). The genome sequencing was then performed by Personal Biotechnology Company (Shanghai, China) by using the Nanopore PromrthION48 platform (Pacific Biosciences platform, CA, USA) and the Illumina Novaseq platform (Illumina, CA, USA).

Raw sequencing data underwent adapter trimming using AdapterRemoval (AdapterRemoval v2) [14], followed by quality filtering and error correction with SOAPec (SOAPdenovo2) [15]. For Illumina short-read data, de novo assembly was performed using SPAdes (SPAdes v3.15.4) [16] and A5-miseq (https://sourceforge.net/projects/ngopt/files/ (accessed on 12 August 2025)) [17]. Nanopore long-read data were assembled via Flye (Flye v2.9.6) [18] and Unicycler (Unicycler v0.5.1) [19] software, with consensus sequences generated through multi-tool integration. Finally, genome polishing achieved 15% N50 improvement using Pilon (Pilon v1.24) [20] through iterative read realignment for base correction and gap closure.

The prediction of genome functional elements encompasses the identification of protein-coding genes and non-coding RNAs. For protein-coding gene prediction, GeneMarkS (http://opal.biology.gatech.edu/GeneMark/genemarks.cgi (accessed on 12 August 2025)) [21] was employed. Non-coding RNAs, including tRNAs, rRNAs, and other regulatory RNAs, were annotated using the Rfam (Rfam 15.0) [22] database, which integrates covariance models and sequence alignments to classify RNA families with high specificity. The functional annotation was performed using BLAST (https://blast.ncbi.nlm.nih.gov/doc/blast-news/2025-BLAST-News.html (accessed on 12 August 2025)) searches against multiple databases, including KEGG (Kyoto Encycolpedia of Gene and Genomes) [23] and COG (Cluster of Orthologous Groups of proteins) [24]. CGView (https://cgview.ca/ (accessed on 12 August 2025)) [25] was employed to provide a comprehensive visualization of the genome features.

### 2.5. Active Compounds Analysis of Hei101

The cryopreserved bacterial strain Hei101 was thawed and aseptically inoculated onto LB agar plates for reactivation. Following 48 h incubation at 30 °C, an isolated colony was transferred to a 300 mL Erlenmeyer flask containing LB broth and cultured under constant agitation (180 rpm) at 30 °C for 48 h. The resultant bacterial suspension was subsequently submitted for specialized analysis of volatile compounds to the Industrial Analysis and Testing Center, Guangdong Academy of Sciences. Volatile organic compounds of bacterial strain Hei101 were collected and analyzed according to the dynamic headspace sampling methodology described by Patrick et al. [26], with modifications to optimize bacterial metabolite capture. For headspace sampling, the system underwent equilibrium conditioning at 60 °C for 30 min under static headspace conditions. Gas chromatography–mass spectrometry (GC-MS) analysis was conducted using a SCION-624MS capillary column (30 m × 0.25 mm × 1.4 μm) with the following precisely controlled temperature program: 1. Initial hold: 50 °C for 3 min; 2. Ramping phase: 6 °C/min to 260 °C; 3. Final hold: 260 °C for 10 min.

### 2.6. Oviposition Selection of BSF

The oviposition apparatus consisted of perforated cylindrical paper shelters (3 × 5 cm) secured to the interior surfaces of an opaque polypropylene chamber (10 × 10 × 8 cm), with one shelter positioned on each vertical face. Each paper substrate received 100 μL aliquots of either compound solution; bacterial suspension (OD600 = 0.8), or sterile water control (*n* = 12 replicates per treatment). Test compounds underwent serial dilution in anhydrous ethanol to achieve working concentrations of 0.3% and 0.03% (*w*/*v*). Adult BSF colonies (*n* = 200 individuals/cage) were introduced into standardized breeding enclosures containing one oviposition device each. Environmental parameters were maintained at 28 °C (±0.5 °C) and 65% relative humidity under a 12:12 h photoperiod (08:00–20:00 light phase). Continuous 72 h oviposition monitoring commenced immediately after device placement, with daily egg mass quantification performed during morning inspections (08:30–09:30).

## 3. Results

### 3.1. Isolation and Identification of Microorganisms from the Surface of BSF Egg

To identify microorganisms that potentially influence BSF oviposition, we initially collected eggs within a specified timeframe post-oviposition and identified six strains from the egg surface (Appendix A). To verify whether microbiota associated with the egg surface originated from the internal microbiota of BSF, we collected eggs under stress-induced oviposition conditions after surface-sterilizing adult females. Only two bacterial strains were confirmed to originate from the maternal microbiota of BSF (Figure 1A). Molecular identification via 16SrDNA sequencing revealed a 99.93% sequence similarity between the two strains, differing by only one base pair (Appendix A). Furthermore, both strains exhibited 99.3% sequence similarity with *Serratia marcescens* KRED strain (GenBank accession number: NR036886.1), confirming their identity as *S. marcescens* (Figure 1B).

### 3.2. Serratia Marcescens Hei101 Attracts the Oviposition of BSF

We examined the oviposition-inducing activity of all the isolated bacteria. The average oviposition quantity of the attracted adults for each strain is presented in Table 1. *Glutamicibacter* sp. Heo1h2404, *S. marcescens* Hei101, and *Microbacterium aurum* Heo2h817 exhibited slight attracting effects compared to the control. However, only *S. marcescens* Hei101 had a significant attracting effect (t = 4.167, *p* < 0.05), and the number of collected egg masses was three times that of the control.

### 3.3. Identification of Active Compounds from Serratia marcescens Hei101

The results of the BSF oviposition induction experiment demonstrated that *S. marcescens* Hei101 had a significant effect on inducing the aggregation and oviposition of BSF. Through headspace solid-phase microextraction and GC-MS analysis, three compounds (Table 2 and Figure 2) were identified from the volatiles of the *S. marcescens* Hei101 bacterial liquid, namely 2-(aziridin-1-yl) ethylamine(1-Azir-EA), acetone, and dimethyl disulfide (DMDS), with 1-Azir-EA being the main component. Acetone, a volatile organic solvent known for its neurotoxicity and mucosal irritation, possesses a distinctly pungent odor that serves as an exposure warning.

Hence, we did not conduct the oviposition attraction experiment with acetone. Subsequently, we, respectively, detected the oviposition attraction effects of DMDS and 1-Azir-EA on BSF (Figure 3). The results indicated that 0.3% DMDS had the best oviposition attraction effect, while 0.03% DMDS had a moderate oviposition attraction ability, but it was still significantly stronger than the bacterial liquid and the control. Regarding 1-Azir-EA, it did not have the effect of attracting BSF to oviposit, as shown in the Figure 3.

### 3.4. Whole-Genome Sequencing and Analysis of Serratia marcescens Hei101

By comparing the genome sequence against the Nucleotide Sequence Database (NT), Hei101 identification was confirmed to be *S. marcescens*, with a sequence similarity of 99.3% to *Serratia marcescens* WVU-009 (GenBank accession number: CP041132.1). The total genome length of *S. marcescens* Hei101 is 5,244,812 base pairs (bp), and the GC content is 59.18% (Figure 4). Gene prediction using Prokka identified 4830 coding sequences (CDS), 90 tRNAs, and 22 rRNAs. Additionally, the genome contains plasmids totaling 74,131 bp in size, with a GC content of 54.23%.

We conducted a BLAST analysis of the genome sequencing results based on the KEGG database, resulting in the annotation of 5486 genes. These genes were categorized into eight major classes and 51 pathways (Figure 5). The genes were primarily enriched in two types of pathways: Brite Hierarchies, which accounted for the largest proportion, and Metabolism, which was the second most represented category. Specifically, 968 genes were enriched in the Protein Families: Signaling and Cellular Processes pathway within Brite Hierarchies, while 707 genes were enriched in the Protein Families: Genetic Information Processing pathway. Additionally, 468 genes were enriched in Carbohydrate Metabolism and 345 genes were enriched in Amino Acid Metabolism.

## 4. Discussion

This is the first report of bacteria directly isolated from the ovaries of BSF. Previous studies have demonstrated that *S. marcescens* can be isolated from larvae and exhibits a notable attraction effect on oviposition [27]. However, recent evidence suggests that *S. marcescens* is not the predominant symbiotic bacterium in BSF larvae [28], possibly due to the antibacterial activity of larval extracts against *S. marcescens* [29,30]. These results corroborate the phenomenon that strain Hei101 is exclusively isolable from the ovarian tissues of BSF but undetectable on the egg surface. Concurrently, these findings imply that strain Hei101 may function as an opportunistic symbiont within BSF.

Insects rely on their sophisticated olfactory systems to detect environmental chemical signals for identifying optimal oviposition sites that ensure offspring survival [31]. Among these cues, microbially derived volatile organic compounds (VOCs) play a pivotal role in insect decision-making processes by providing critical information about host nutritional quality, predator risks, and interspecific/intraspecific competition pressures [32,33,34]. Our experimental results revealed that *S. marcescens* Hei101 isolated from BSF ovaries produced three volatile compounds during fermentation, with dimethyl disulfide (DMDS) demonstrating exclusive oviposition attraction to gravid BSF females. This suggests that DMDS may serve as a key semiochemical enabling BSF to assess environmental suitability for larval development.

The ecological significance of DMDS in insect oviposition behavior has been extensively documented across diverse taxa, including *Anopheles coluzzii* [35], *Drosophila busckii* [36], *Anopheles gambiae* [37]. However, research focusing on DMDS to BSF remains scarce. Michishita et al. [38] recently demonstrated BSF larvae’s capacity to significantly reduce environmental DMDS concentration. When combined with our findings, this raises the possibility that BSF larvae may actively metabolize Hei101-derived DMDS during their developmental process. Such a mechanism could represent an evolutionary adaptation to simultaneously exploit microbial resources while modifying chemical cues to optimize offspring habitat selection—a hypothesis warranting further investigation through microbial community dynamics analysis and isotope tracing techniques.

## 5. Conclusions

This study demonstrates that a *S. marcescens* strain Hei101 from the ovaries of BSF can attract oviposition of BSF while the other six bacterial strains from the egg surface cannot. The Hei101 strain releases three volatile compounds: 1-Azir-EA, acetone, and DMDS. DMDS exhibited significant oviposition-attractant properties toward gravid BSF females (*p* < 0.01), while 1-Azir-EA showed no influence of oviposition. Genomic characterization revealed the Hei101 strain possesses a 5,244,812 bp chromosome with 59.18% GC content, encoding 5486 predicted genes as annotated via KEGG pathway analysis. Collectively, this work provides a theoretical foundation for enhancing BSF production efficiency and offers insights into further microbial regulation development.

## Figures and Tables

**Figure 1 biology-14-01107-f001:**
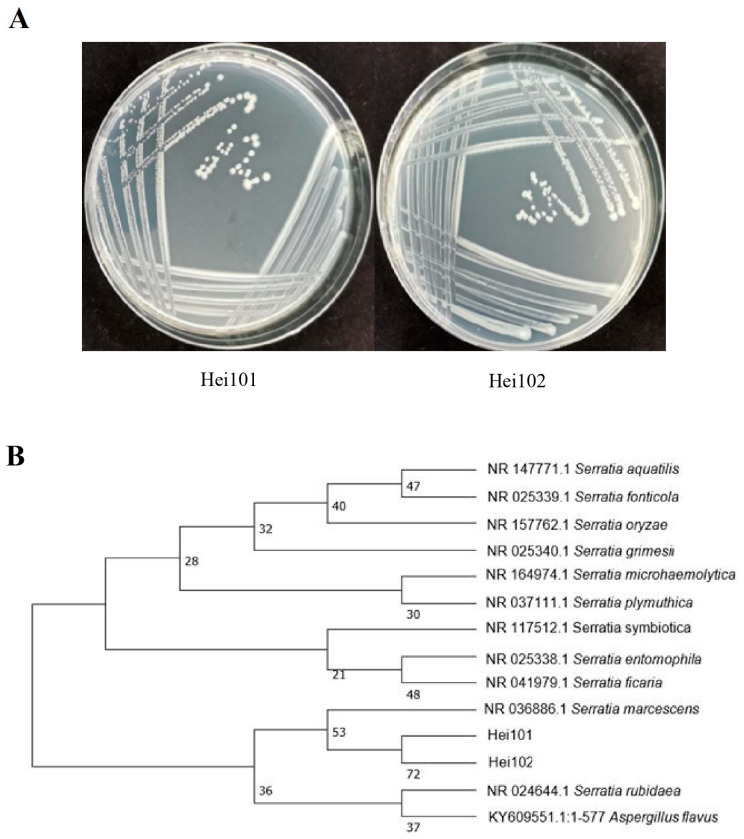
**The identification of Serratia marcescens Hei101 and Hei102**. (**A**) Colonial morphology of *Serratia marcescens* strains Hei101 and Hei102. (**B**) Phylogenetic tree reconstructed using the Neighbor-Joining method based on 16S rRNA gene sequences.

**Figure 2 biology-14-01107-f002:**
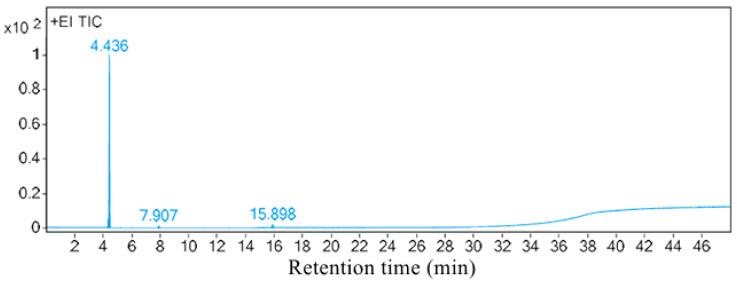
Peak map of volatiles from *Serratia marcescens* Hei101.+EI TIC: Total Ion Chromatogram (TIC) in electron ionization (EI) mode.

**Figure 3 biology-14-01107-f003:**
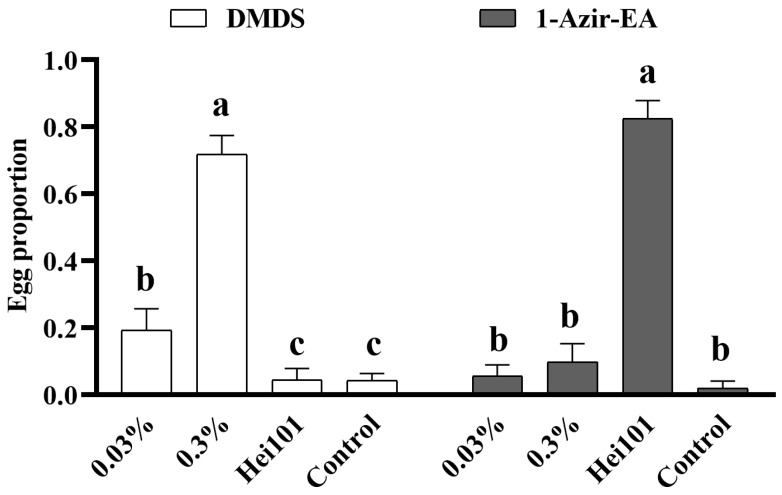
**Oviposition attractant effects of two volatiles from Hei101 on BSF.** The bar chart illustrates the efficacy of DMDS or 1-Azir-EA in attracting black soldier fly (BSF) oviposition. Horizontal axis labels denote the mass fraction concentrations of each compound (0.03% and 0.3%), with experimental groups including Hei101 bacterial suspension (OD_600_ = 0.8) and a sterile water control. The same letters above the error bars indicate no significant differences at the 0.05 level with the Kendall nonparametric test (*n* = 12).

**Figure 4 biology-14-01107-f004:**
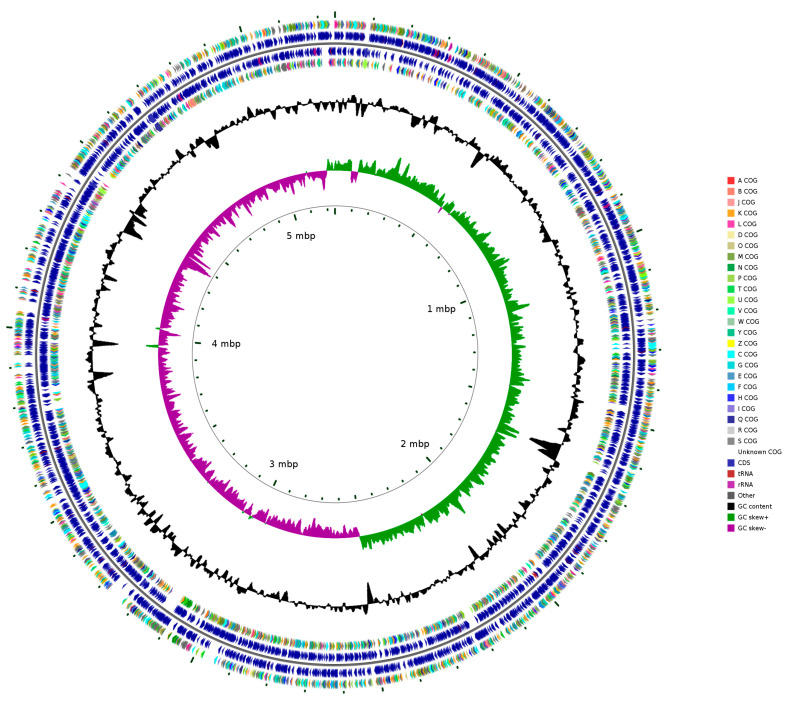
**The genomic map of *Serratia marcescens* Hei101.** The circular diagram features seven concentric rings from the innermost to the outermost: the first ring displays the genomic scale markers; the second ring illustrates GC skew values; the third ring represents GC content distribution; the fourth and seventh rings depict Clusters of Orthologous Groups (COG) classifications for each coding sequence (CDS); the fifth and sixth rings annotate the genomic positions of CDSs, tRNAs, and rRNAs, with distinct color-coding to differentiate these functional elements.

**Figure 5 biology-14-01107-f005:**
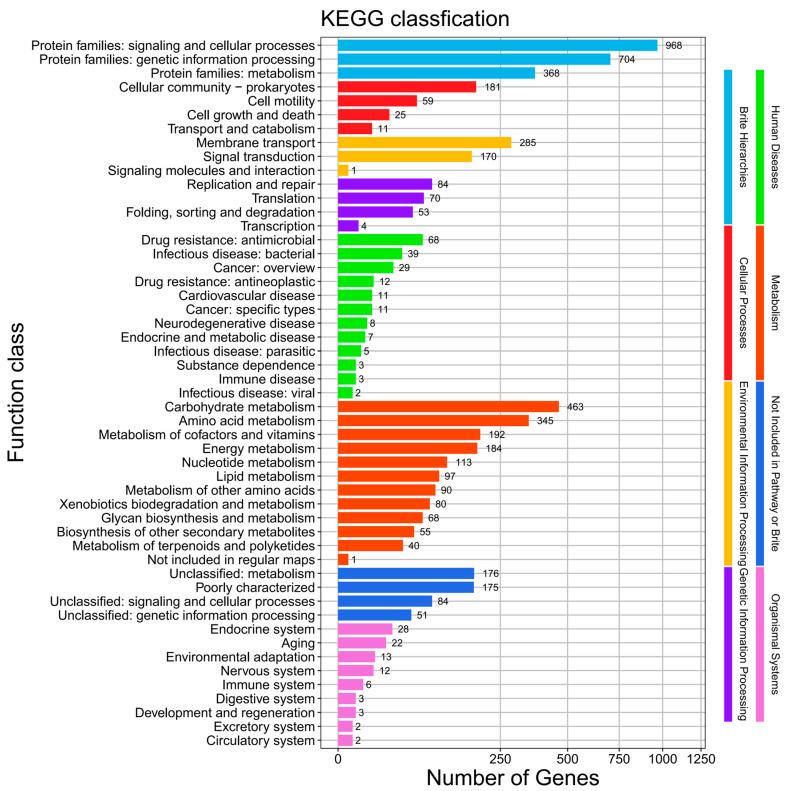
KEGG functional classification of Serratia marcescens Hei101.

**Table 1 biology-14-01107-t001:** **Numbers of egg clusters collected from egg-collecting boxes treated by different bacteria.** The data in the same row of the table indicate the egg-laying attraction effect of the corresponding bacteria on *Hermetia illucens*, and each experiment was repeated 6 times. Data were analyzed by independent sample Student’s T test.

Strain	No. of Egg Clusters		*p* Value
	Treatment	Control	
*Cellulosimicrobium* sp. Heo2h808	0.5 ± 0.22	1.17 ± 0.4	0.286
*Glutamicibacter* sp. Heo1h2404	4.83 ± 1.22	2.17 ± 0.65	0.203
*Stahphylococcus xylosus* Heo2h2404	0.83 ± 0.54	2.67 ± 0.99	0.247
*Serratia marcescens* Hei101	5.88 ± 1.25	1.75 ± 0.45	0.004
*Rhizobium* sp. Heo2h816	0.25 ± 0.16	0.25 ± 0.16	1.000
*Bacillus amyloliquefaciens* Heo1h801	1.13 ± 0.67	1.13 ± 0.51	1.000
*Microbacterium aurum* Heo2h817	1.67 ± 0.42	0.17 ± 0.17	0.541

**Table 2 biology-14-01107-t002:** Volatile substances from culture broth of *Serratia marcescens* Hei101.

Peak	Retention Time /min	Matching Degree /%	Chemical Formula	Peak Area	Area Percentage /%	CAS	Compound
1	4.438	84.0	C4H10N2	298345000	96.58	4025-37-0	2-(Aziridin-1-yl)ethan-1-amine
2	7.912	87.3	C3H6O	425148	1.37	67-64-1	acetone
3	15.898	88.0	C2H6S2	6331841	2.05	624-92-0	dimethyl disulfide

## Data Availability

The original datasets and genomic data generated in this study are available in the NCBI database under accession number SAMN49980336. The other original contributions presented in this study are included in the article/Appendix A. Further inquiries may be directed to the corresponding authors.

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
