# Peer review of "A Bacterium Derived from the Ovary of the Black Soldier Fly (Hermetia illucens) Attract Oviposition of the Host"

_biology, 2025, doi:10.3390/biology14091107_

Round 1

Reviewer 1 Report

Comments and Suggestions for Authors

‘A bacterium derived from the ovary of the black soldier fly (Hermetia illucens) attract oviposition of the host’ promises important contributions to the scientific community, particularly in addressing food insecurity through insect protein production. However, I have some observations regarding the quality of the manuscript in its current form. These are stated below:

  1. Abstract: The aim(s)/objective(s) of the study are not clearly stated. This should be addressed to guide the reader's understanding and provide focus to the study.
  2. References: The in-text reference style does not conform to the journal’s required style. Please revise it accordingly.
  3. Introduction: While the knowledge gaps are well articulated, the specific objectives of the study are missing. Including them will provide a clear framework and focus for the research.
  4. Methodological Placement: Methodological details currently included in the Introduction should be relocated to the Materials and Methods section, as per standard manuscript structure.
  5. Feed Specification: The specific type of chicken feed used should be clearly stated, considering the various classes and formulations available.
  6. Feeding Consistency: Please clarify whether the same feed was administered throughout the experimental period. This information is essential for reproducibility and understanding of the nutritional regimen.
  7. Honey Supplementation: The exact quantity of honey added to the adult diet should be clearly specified to ensure clarity and reproducibility.
  8. Formatting and Style: The manuscript should be revised to conform fully to the journal’s formatting and presentation style.
  9. Line 162: Providing further explanation on the rationale for stress-induced oviposition conditions would enhance clarity and aid reader comprehension.
  10. References in Methods: While the methodology appears to follow standard procedures, only a limited number of references are cited. Additional relevant references should be included to validate the protocols used and properly acknowledge the original sources.
  11. Line 165: Please clarify the basis or rationale for selecting the stated incubation temperatures.
  12. Methodological Details in Results: Any methodological or rationale-related information currently placed in the Results section should be moved to the Materials and Methods
  13. Repetition (Lines 74–76 and 119–121): These sections appear repetitive and should be reviewed. The repetition is unnecessary and should be removed or rephrased for clarity.
  14. Scientific Names (e.g., Line 250): Scientific names should be italicized and written in full (Genus species) on first mention, then abbreviated appropriately thereafter. Please ensure consistency throughout the manuscript.
  15. Discussion: The Discussion section should be expanded to thoroughly interpret the results presented. Deeper analysis and comparison with findings from related studies are recommended.

Author Response

Dear Reviewers:

Thank you for your letter and for the reviewer's comments concerning our manuscript entitled“A bacterium derived from the ovary of the black soldier fly (Hermetia illucens) attract oviposition of the host”(ID: biology-3795292).Those constructive comments are all valuable and very helpful for revising and improving our paper, as well as the important guiding significance to our researches. We have studied comments carefully and have made correction which we hope meet with approval. Highlighting function is used for all corrections in the manuscript. We replied each comment respectively below.

Comments 1: Abstract: The aim(s)/objective(s) of the study are not clearly stated. This should be addressed to guide the reader's understanding and provide focus to the study.

Response 1: The abstract has been revised as follows. “ The black soldier fly, BSF (Hermetia illucens) has extensive applications in insect protein production and organic waste conversion, serving as a crucial resource insect. However, large-scale breeding faces challenges such as low adult mating rates, unstable oviposition, and inefficient egg collection, which significantly hinder the industrial application of BSF. In this study, we aimed to enhance the oviposition efficiency of BSF by utilizing the microbes within it. We isolated a strain of Serratia marcescens from the ovaries of the BSF, which can attract BSF to lay eggs by producing dimethyl disulfide. Genome analysis of the bacterium revealed a total length of 5,244,812 bp with a GC content of 59.18%. Based on KEGG database annotations, 5,486 genes were identified through genome sequencing. The findings of this study provide a theoretical foundation for enhancing BSF production efficiency and offer insights for further microbial regulation development.”

Comments 2: References: The in-text reference style does not conform to the journal’s required style. Please revise it accordingly.

Response 2: Done.

Comments 3: Introduction: While the knowledge gaps are well articulated, the specific objectives of the study are missing. Including them will provide a clear framework and focus for the research.

Response 3: We have revised the final paragraph of the introduction section as follows. “In this study, we sought to improve the oviposition efficiency of BSF via utilization of its microbiota. we isolated a symbiotic bacterium from the ovaries of BSF that exhibits a strong attraction for BSF oviposition. The active odorant compounds responsible for this attraction were identified through gas chromatography-mass spectrometry (GC-MS) analysis and behavioral assays. Additionally, genome sequencing of the strain was performed to delineate its genetic repertoire and characterize functional potential through comprehensive bioinformatic annotation.”

Comments 4: Methodological Placement: Methodological details currently included in the Introduction should be relocated to the Materials and Methods section, as per standard manuscript structure.

Response 4: After careful review of the Introduction, we found no methodological details as referenced in the review. We kindly request the reviewer to specify the exact location where such content is expected to facilitate targeted revision. 

Comments 5: Feed Specification: The specific type of chicken feed used should be clearly stated, considering the various classes and formulations available.

Response 5: Detailed specifications of the chicken feed (including supplier and composition) has now been added to the Materials And Methods section." See line 77.

Comments 6: Feeding Consistency: Please clarify whether the same feed was administered throughout the experimental period. This information is essential for reproducibility and understanding of the nutritional regimen.

Response 6: Identical feed was used throughout the experiment. As requested, this information were added to the Materials And Methods section. See line 86

Comments 7: Honey Supplementation: The exact quantity of honey added to the adult diet should be clearly specified to ensure clarity and reproducibility.

Response 7: The concentration of the honey water has been documented in the Materials And Methods section. See line 83

Comments 8: Formatting and Style: The manuscript should be revised to conform fully to the journal’s formatting and presentation style.

Response 8: Done.

Comments 9: Line 162: Providing further explanation on the rationale for stress-induced oviposition conditions would enhance clarity and aid reader comprehension.

Response 9: Full procedural descriptions are now documented in the Materials And Methods. See line 88-96.

Comments 10: References in Methods: While the methodology appears to follow standard procedures, only a limited number of references are cited. Additional relevant references should be included to validate the protocols used and properly acknowledge the original sources.

Response 10: The relevant references of BSF rearing and volatile compounds identification have been added. See line 73 and line 166.

Comments 11: Line 165: Please clarify the basis or rationale for selecting the stated incubation temperatures.

Response 11: Selective microbial isolation was achieved through differential temperature incubation:26℃ and 30℃ (approximating BSF adult rearing conditions); 37℃ (to exclude environmental/human-accociated contaminants).

Comments 12: Methodological Details in Results: Any metdological or rationale-related information currently placed in the Results section should be moved to the Materials and Methods

Response 12: We have thoroughly reviewed the Results section but were unable to locate the descriptions referenced. Could you kindly specify where this content should be integrated to ensure precise revision.

Comments 13: Repetition (Lines 74–76 and 119–121): These sections appear repetitive and should be reviewed. The repetition is unnecessary and should be removed or rephrased for clarity.

Response 13: We contend these sections are methodologically distinct. Lines 74-76 describes preliminary identification via 16SrDNA fragment alignment, whereas lines 119-121 involves comprehensive genomic sequence identification.

Comments 14:Scientific Names (e.g., Line 250): Scientific names should be italicized and written in full (Genus species) on first mention, then abbreviated appropriately thereafter. Please ensure consistency throughout the manuscript.

Response 14: We apologize for overlooking the scientific name writing formation in the manuscript. The entire text has now been thoroughly reviewed and corrected.

Comments 15: Discussion: The Discussion section should be expanded to thoroughly interpret the results presented. Deeper analysis and comparison with findings from related studies are recommended.

Response 15: We maintain that the Discussion section has thoroughly discuss the key findings. To further enhance the manuscript, could you kindly specify which particular results require deeper analysis? We gratefully acknowledge your guidance.

Reviewer 2 Report

Comments and Suggestions for Authors

This paper describes isolation of Serratia marcescens, an extremely widespread and cosmopolitan strain, in the ovary of the BSF, and found it attracts BSF through production of dimethyl disulfide.

A major comment is that the methods are missing the description of the forced oviposition assay. How did you extract bacteria from the ovaries ensuring there was no contamination? This is a major part of the study, so these methods must be described with enough detail to be replicated.

Some text is very clearly AI-written and must be deleted.

Small comments
23-33 This can be deleted if you want. Everyone knows about BSF now.
42 "these insects" is vague. Is it BSF, or the competing insects? Why would bacteria isolated from insects competing with BSF inhibit egg production of the insects that are the source of the bacteria?
42-44 You say "current research," but you only cited one paper (Zheng et al. 2013). I'd like to see more citations of past research on microbes affecting oviposition in BSF.
45-50 Delete. Talk only about microbes as oviposition cues and delete anything broader.
60 Symbiotic is technically true, but in the ears of many it will sound like "mutualistic." This could be a pathogen, as Serratia marcescens often in. Delete "symbiotic" to be safe.
245-252 I ran this text through an AI detector and got 100% AI written. The word "elucidating" is a dead giveaway. All this text is worthless anyway, so delete it all, and start the Discussion with "This is the first report of bacteria…"
256 Italics for Serratia marcescens
257 "These findings highlight the diverse interspecific interactions involving Serratia marcescens." Do they, or is this another AI-derived sentence? 
270-280 Delete "notably" "Intriguingly" "intriguing"

Author Response

Dear Reviewers:

Thank you for your letter and for the reviewer's comments concerning our manuscript entitled“A bacterium derived from the ovary of the black soldier fly (Hermetia illucens) attract oviposition of the host”(ID: biology-3795292).Those constructive comments are all valuable and very helpful for revising and improving our paper, as well as the important guiding significance to our researches. We have studied comments carefully and have made correction which we hope meet with approval. Highlighting function is used for all corrections in the manuscript. We replied each comment respectively below.

Comments 1: A major comment is that the methods are missing the description of the forced oviposition assay. How did you extract bacteria from the ovaries ensuring there was no contamination? This is a major part of the study, so these methods must be described with enough detail to be replicated.

Response 1: Full procedural descriptions are now documented in the Materials And Methods. See line 88-96.

Comments 2: 23-33 This can be deleted if you want. Everyone knows about BSF now.

Response 2: We consider that an introductory overview of the BSF is indispensable for non-entomology researchers. Therefore, we decide to retain the content.

Comments 3: 42 "these insects" is vague. Is it BSF, or the competing insects? Why would bacteria isolated from insects competing with BSF inhibit egg production of the insects that are the source of the bacteria?

Response 3: The sentence has been revised. See line 52.

Comments 4: You say "current research," but you only cited one paper (Zheng et al. 2013). I'd like to see more citations of past research on microbes affecting oviposition in BSF.

Response 4: To our knowledge, only one relevant published SCI study was identified. Consequently, the original text has been revised to eliminate potential misinterpretation. See line 52.

Comments 6: 45-50 Delete. Talk only about microbes as oviposition cues and delete anything broader.

Response 6: Done.

Comments 7: 60 Symbiotic is technically true, but in the ears of many it will sound like "mutualistic." This could be a pathogen, as Serratia marcescens often in. Delete "symbiotic" to be safe.

Response 7: Done.

Comments 8: 245-252 I ran this text through an AI detector and got 100% AI written. The word "elucidating" is a dead giveaway. All this text is worthless anyway, so delete it all, and start the Discussion with "This is the first report of bacteria…".

Response 8: The relevant content has been rewritten. See line 266-273.

Comments 9: Italics for Serratia marcescens

Response 9: Done.

Comments 10: 257 "These findings highlight the diverse interspecific interactions involving Serratia marcescens." Do they, or is this another AI-derived sentence?.

Response 10: All references cited are authentic scholarly sources. Relevant paragraph has been rewritten. See line 266-273.

Comments 11: Delete "notably" "Intriguingly" "intriguing"

Response 11: Done.